# NPC1 Confers Metabolic Flexibility in Triple Negative Breast Cancer

**DOI:** 10.3390/cancers14143543

**Published:** 2022-07-21

**Authors:** Kathleen I. O’Neill, Li-Wei Kuo, Michelle M. Williams, Hanne Lind, Lyndsey S. Crump, Nia G. Hammond, Nicole S. Spoelstra, M. Cecilia Caino, Jennifer K. Richer

**Affiliations:** 1Department of Pathology, University of Colorado Anschutz Medical Campus, Aurora, CO 80045, USA; kathleen.oneill@cuanschutz.edu (K.I.O.); li-wei.kuo@cuanschutz.edu (L.-W.K.); michelle.m.williams@cuanschutz.edu (M.M.W.); hanne.lind@cuanschutz.edu (H.L.); lyndsey.crump@cuanschutz.edu (L.S.C.); niahammond@uchicago.edu (N.G.H.); nicole.spoelstra@cuanschutz.edu (N.S.S.); 2Department of Pharmacology, University of Colorado Anschutz Medical Campus, Aurora, CO 80045, USA; cecilia.caino@cuanschutz.edu

**Keywords:** breast cancer, cholesterol, NPC1, mitochondria

## Abstract

**Simple Summary:**

Triple negative breast cancer is an aggressive breast cancer subtype with limited targeted therapeutic options. As a method of identifying novel therapeutic targets in this disease subtype, we utilize a microRNA that reverses the Epithelial-to-Mesenchymal Transition, which reveals Niemann-Pick C1 (*NPC1*) as a gene highly expressed in triple negative breast cancer. Silencing of NPC1 causes significant loss of tumor-promoting capabilities of these cell lines. We find that NPC1 promotes cell proliferation in soft agar and invasive capacity, while silencing impairs these functions and leads to mitochondrial dysfunction and suppression of pro-tumorigenic signaling. This work suggests NPC1 as a potential target and a mediator of breast cancer aggression.

**Abstract:**

Triple-negative breast cancer (TNBC) often undergoes at least partial epithelial-to-mesenchymal transition (EMT) to facilitate metastasis. Identifying EMT-associated characteristics can reveal novel dependencies that may serve as therapeutic vulnerabilities in this aggressive breast cancer subtype. We found that *NPC1*, which encodes the lysosomal cholesterol transporter Niemann–Pick type C1 is highly expressed in TNBC as compared to estrogen receptor-positive (ER+) breast cancer, and is significantly elevated in high-grade disease. We demonstrated that *NPC1* is directly targeted by microRNA-200c (miR-200c), a potent suppressor of EMT, providing a mechanism for its differential expression in breast cancer subtypes. The silencing of *NPC1* in TNBC causes an accumulation of cholesterol-filled lysosomes, and drives decreased growth in soft agar and invasive capacity. Conversely, overexpression of NPC1 in an ER+ cell line increases invasion and growth in soft agar. We further identified TNBC cell lines as cholesterol auxotrophs, however, they do not solely depend on NPC1 for adequate cholesterol supply. The silencing of *NPC1* in TNBC cell lines led to altered mitochondrial function and morphology, suppression of mTOR signaling, and accumulation of autophagosomes. A small molecule inhibitor of NPC1, U18666A, decreased TNBC proliferation and synergized with the chemotherapeutic drug, paclitaxel. This work suggests that NPC1 promotes aggressive characteristics in TNBC, and identifies NPC1 as a potential therapeutic target.

## 1. Introduction

Triple-negative breast cancer (TNBC) is an aggressive subtype representing 15% of newly diagnosed breast cancer cases. Identification of targetable vulnerabilities is imperative because TNBC often acquires resistance to chemotherapy (30–50% of patients) [1]. Compared to estrogen-receptor positive (ER+) and human epidermal growth factor amplified (HER2+) breast cancers, TNBCs have a rapid peak rate of recurrence as metastatic disease, often within 3–5 years of initial diagnosis [2]. TNBCs have significantly lower expression of the microRNA miR-200 family than ER+ BC and normal breast epithelium [3,4]. This miRNA family, known as the “guardian of the epithelial phenotype”, potently suppresses epithelial-to-mesenchymal transition (EMT) [3,5,6,7]. The silencing or deletion of miR-200c by methylation or micro-deletions allows EMT to occur by permitting the translation of developmental, mesenchymal, and neuronal genes not typically expressed in normal epithelial cells [8,9,10,11]. Work with human TNBC cell lines and mouse mammary carcinoma models demonstrated that restoration of miR-200c to TNBC drives a switch to a more epithelial, less invasive, and less metastatic phenotype, through direct targeting of mesenchymal transcription factors including ZEB1, a suppressor of E-cadherin, and other epithelial markers. Restoration of miR-200c to TNBC can, thus, reveal potential dependencies of TNBC. 

In addition to repressing genes traditionally associated with EMT, restoration of miR-200c to TNBC cells reduces levels of immune-modulatory and metabolic genes [12,13]. We discovered that *NPC1*, encoding the lysosomal cholesterol transporter Niemann–Pick type C-1, was among the genes most significantly reduced by miR-200c, along with additional changes to genes involved in cholesterol and lipid metabolism [12], suggesting cholesterol transport as a potential pathway of interest in TNBC progression. Cellular cholesterol is tightly regulated, and can be derived from de novo biosynthesis or lysosome-mediated uptake. Following endocytosis of cholesterol-containing low-density lipoproteins (LDL) from the microenvironment, cells process cholesterol within the lysosome, and utilize NPC1 to efflux cholesterol to the endoplasmic reticulum [14]. NPC1 is best studied in the context of loss-of-function mutations that led to the Niemann–Pick type C lysosomal storage disorder, characterized by abnormal accumulation of cholesterol within the lysosome, driving trafficking defects, and lysosomal dysfunction [15,16,17]. NPC1 is also the mechanism by which mTOR senses cholesterol levels, and serves as a component of the lysosomal mTOR signaling scaffold [18,19].

In the context of cancer, NPC1 activity and function are understudied. Here, we report that NPC1 may serve as a potential therapeutic target in TNBC, where it is elevated compared to ER+ BC. We find that restoration of miR-200c directly targets *NPC1* and represses NPC1 protein levels. The suppression of NPC1 in TNBC leads to slowed proliferation, decreased anchorage-independent survival and decreased invasion, and altered mitochondrial function. Thus, NPC1 may serve as a potential therapeutic target in TNBC.

## 2. Methods

### 2.1. Cell Culture and Reagents

MDA-MB-231 and BT549 cell lines were purchased directly from ATCC (Manassas, VA, USA). Sum159PT (RRID:CVCL_5423) cells were purchased from the University of Colorado Cancer Center Tissue Culture Core (Aurora, CO, USA), and the MCF7 cells were obtained from Dr. Kate Horowitz at the University of Colorado Anschutz Medical Campus. Cell lines were authenticated by short tandem repeat DNA profiling (Promega, Madison, WI, USA) and tested for mycoplasma at the University of Colorado Cancer Center (UCCC) Cell Technologies Shared Resource (September 2020). For culture, MDA-MB-231 (MDA-231) cells were grown in MEM with 5% FBS, 1 mM HEPES, 2 mM L-glutamine, and insulin; BT549 (NCI-DTP Cat# BT-549, RRID: CVCL_1092) cells were cultured in RPMI 1640 medium with 10% FBS, and insulin; MCF7 in MEM with 5% FBS and insulin; Sum159PT cells were grown in Ham F-12 with 5% FBS, penicillin/streptomycin, hydrocortisone, insulin, HEPES, and L-glutamine supplementation. Prior to experimentation, all cells were switched to DMEM + 10% FBS, which was used throughout all experiments unless otherwise noted.

#### Stable NPC1 Knockdown and Exogenous Expression Cell Lines

pLVX-NPC1(WT)-FLAG (RRID: Addgene_164972) was a gift from R. Zoncu at the University of California, Berkeley. pcDNA3.1 (RRID: Addgene_79663) and pcDNA3.1-NPC1 were from Genscript (Piscataway, NJ, USA). pLKO-empty and pLKO encoding NPC1-targeted shRNAs were purchased from The University of Colorado-Anschutz Functional Genomics Core (Aurora, CO, USA), with shRNA #1 targeted to TRCN0000005428 (3’UTR), and shRNA #2 targeted to TRCN0000418552 (CDS). Cells were selected in neomycin (500 µg/mL) or puromycin (1 µg/mL) for ~2 weeks prior to experimentation.

### 2.2. Transfections

*miR-200c experiments:* Negative, scrambled control (4464058, Thermofisher Scientific, Waltham, MA, USA) or miR-200c (4464066, MC11714, Thermofisher Scientific, Waltham, MA, USA) mimics were used at a final concentration of 50 nmol/L. All transfections were performed using either RNAi Max or Lipofectamine 3000 (Thermo Fisher Scientific) and experiments were conducted following the manufacturer’s protocols.

*Luciferase assay:* 50 nM of scramble mimic negative control or miR-200c-3p mimic was co-transfected with 1 μg of Pmir-glo (E1330, Promega) fused with NPC1-3’UTR containing seed regions of miR-200c binding site or mutant. Cells were lysed after 48 h of transfection. The luciferase activities were measured by the Dual-Luciferase^®^ Reporter Assay system (E1910, Promega) according to the manufacture instruction with internal Renilla control. Site-direct mutagenesis of the seed region was performed utilizing GeneArt™ Site-Directed Mutagenesis PLUS system (A14604, Invitrogen, Waltham, MA, USA). The primers used are listed in Appendix A.

*NPC1 siRNA silencing:* Transient siRNA transfections were performed using 30–60 µM NPC1 siRNA from Ambion/Thermofisher (catalog AM16704, ID 106016), in combination with Lipofectamine RNAi Max (ThermoFisher) and conducted according to the manufacturer’s protocols. 

### 2.3. qRT-PCR

Total RNA was isolated using TRIzol RNA Isolation (Qiagen, Hilden, Germany) and cDNA was synthesized with the qScript cDNA SuperMix (QuantaBio/Qiagen, Hilden, Germany). qRT-PCR was performed on an ABI 7600 FAST thermal cycler. SYBR Green quantitative gene expression analysis was performed using ABsolute Blue qRT-PCR SYBR Green Low ROX Mix (Thermo Fisher Scientific). Primers are listed in Appendix A.

### 2.4. Immunoblotting

Whole-cell protein extracts (20–40 μg) were denatured using RIPA buffer, separated on SDS-PAGE gels, and transferred to polyvinylidene fluoride membranes. After blocking in 5% non-fat milk in Tris-buffered saline–Tween, membranes were probed overnight at 4 °C. The following antibodies were used in this study: NPC1 (Novus Biologicals, Parker, CO, USA, NB400); LDLR (Novus Biologicals, NBP1-06709); HMGCR (ThermoFisher PA5-37367); HMGCS1 (Thermo Fisher Scientific Cat# PA5-29488, RRID:AB_2546964); DHCR24 (Cell Signaling Technology Cat# 2033, RRID:AB_2091448); LAMP1 (Abcam, Cambridge, United Kingdom, Cat# ab25630, RRID: AB_470708); Phospho-S6 Kinase (Thr389) (Cell Signaling, #9205); SQSTM1/P62 (Cell Signaling #5114); LC3B (Cell Signaling, Danvers, MA #3868). Secondary antibodies were Licor IR Dye goat anti-rabbit 800 IgG (#926-32211) or goat anti-mouse 680 (#926-68070).

### 2.5. Nutrient Deprivation

Lipoprotein-depleted fetal bovine serum (LPDS, Kalen Biomedical, Montgomery, MD, USA, #880100) used at 5% or 10% in DMEM, and when noted, supplemented with 10 µg/mL of human LDL (Kalen Biomedical 770200-4). For MBC: Cholesterol complexing, methyl-β-cyclodextrin (Medchem Express, Monmouth Junction, NJ, USA, HY-101461) and cholesterol (Sigma, St. Louis, MO, USA, C8667) were warmed to 67 °C, combined at a 1:1 molar ratio (1% MBC to 20 µg/mL Chol), added to 37 °C media, and vortexed. For amino acid starvation, EBSS was supplemented with 5 mM glucose, 1 mM sodium pyruvate, and 2 mM glutamine. 

### 2.6. Dil-LDL Uptake

Cells seeded 24–48 h in 24-well dishes were washed 1× with PBS and starved in 5% LPDS in DMEM for 1 h, then given 10 µg/mL dil-LDL (Kalen Biomedical # 770230-9) in 5% LPDS/DMEM and incubated for ~5 h at 37 °C. Following lysis in RIPA, cells were spun for 5 min at 20,000 g. A total of 5 µL of supernatant was transferred to a 96-well plate reader (Biotek, Winooski, VT, USA) and read at 520/580 nm. The remaining supernatant was used to measure protein with a BCA assay, and uptake was normalized to µg of protein.

### 2.7. Two-Dimensional and Three-Dimensional Growth and Invasion Assays

For U18666A IC-50 measurements, BT549, MDA-MB-231, and Sum159PT cells were seeded in 96-well dishes at 6k, 5k, and 3k cells per well, respectively. Cells were given serial dilutions of U18666A for 48 h and compared to 1% DMSO control. Cells were fixed in 10% neutral buffered formalin for 20 min, washed with PBS, and incubated in crystal violet stain for 20 min before rinsing with diH_2_O. Plates were dried overnight and the stain was resuspended in 5% acetic acid, absorbance was measured at 570 nm. 

For U18666A and Paclitaxel combination, BT549 cells were seeded in 96-well dishes at 6k cells/well. The next day, 3 doses of paclitaxel (2, 7, 12 nM) and 3 doses of U18666A (2, 4, 8 µM) were given alone or in combination with n = 3 wells each. After 48 h, crystal violet was performed. Additive/synergy calculations were performed using SynergyFinder 2.0 (SynergyFinder, RRID:SCR_019318).

Confluency over time of NPC1 siRNA cells was conducted using xCELLigence RTCA (Agilent Technologies, Santa Clara, CA). Cells were seeded in xCELLigence plates 24 h following transfection and “cell index” was measured every 60 min for 96 h.

For 3D/soft agar assays, cells were plated in triplicates in 6-well plates in 0.5% bottom agar and 0.3% top agar containing 5000 cells/well (Sum159PT) or 30,000 cells/well (MCF7). Cells were grown for approximately 14 days with a once-weekly media change and stained with nitro blue tetrazolium at harvest. For LPDS soft agar experiments, agar contained 5% LPDS-DMEM, with or without 10 µg/mL LDL and these concentrations were maintained in top media. For MBC:Cholesterol experiments, 5% FBS-DMEM was given vehicle or 20 µg/mL cholesterol in a 1:1 molar ratio complex with MBC, as described previously. Experiments were performed three separate times, imaged using Optronix GelCount, and quantified using ImageJ software (ImageJ, RRID:SCR_003070).

### 2.8. Invasion Assay

Three-dimensional transwell invasion assays were performed with 8.0 µm pore transwell inserts in 24-well plates. Filters were coated overnight with 200 µg/mL Cultrex UltiMatrix Basement Membrane Extract (R&D Systems, Minneapolis, MN, USA, Cat. # BME001). A total of 50,000 cells per well were then seeded in serum-free media. Invasive cells were assessed 24 h later via fixation with 10% NBF and staining with 0.1% crystal violet. 

### 2.9. Microscopy

#### Mitochondrial Immunocytochemistry and Quantification

Cells were grown on glass coverslips for 48 h, then fixed in 10% formalin (Thermo Fisher Scientific Cat*#*
*SF93*), permeabilized with 0.1% Triton-X100, and blocked in 10% normal goat serum with 0.3 M glycine in PBS. Fixed cells were incubated overnight with mouse anti-mitochondria (MTC02, Thermo Fisher Scientific Cat# MA5-12017, RRID:AB_10983622) diluted in a 1:500, washed in PBS 3 times, followed by 45 min at room temperature with anti-mouse Alexa Fluor 568 secondary antibody (Thermo Fisher Scientific Cat# A-11004, RRID:AB_2534072) diluted in a 1:500, and slides were mounted using Prolong Diamond Antifade Mountant DAPI (Thermo Fisher P36966). For qualitative scoring of cells according to mitochondrial morphology, imaging was created using an Olympus BX40 microscope with a 100x/objective lens and FITC filter.

Random fields (25–30) were analyzed, and each cell was classified into 3 categories according to the main mitochondrial morphology (small, medium, or elongated). For automated analysis of mitochondrial size, slides were scanned in a Zeiss LSM780 confocal microscope (Carl Zeiss LLC, United States) with a 63× oil objective. Z-stack from 5 random fields was acquired with a slice thickness of 0.45 microns, and the built-in 3D function on the Zeiss software was used to render 3D images of the mitochondrial network surface at 1,4, and 8× zoom. Images were analyzed in ImageJ (RRID:SCR_003070) to obtain the average mitochondrial size per cell, from ~20 cells per condition.

### 2.10. Filipin Stain

Cells were grown on glass coverslips, and fixed in 10% formalin, quenched with 100 mM glycine. A total of 0.05 mg/mL filipin (Santa Cruz, Dallas, TX, USA, #480-49-9) was used in 10% normalized goat serum in PBS and stained for 1.5 h, mounted with Prolong Diamond Antifade Mountant, and imaged using an Olympus BX40 microscope and a 40× objective.

### 2.11. NPC1 IHC

Formalin-fixed paraffin-embedded tissues were cut at 5 µm, baked, and deparaffinized in a series of xylenes and graded ethanols. Antigens were heat retrieved using citrate buffer, and tissues were stained for NPC1 using Novus Bio #NB400-148 primary antibody followed by goat anti-rabbit HRP, and DAB (Vector Labs, RRID:AB_2819346).

### 2.12. Mitochondrial Assay

The Seahorse Mitochondrial stress test was performed according to the manufacturer’s instructions using the Seahorse XFe96 format (Agilent). On the morning of the experiment, cells and media were changed to Seahorse XF DMEM medium, pH 7.4 (103575-100) with 10 mM glucose, 2 mM glutamine, and 1 mM sodium pyruvate. Media was also refreshed immediately prior to the assay run. Basal oxygen consumption rate (OCR) was measured prior to the addition of mitochondrial inhibitors: Oligomycin 2 µM, carbonilcyanide p-triflouromethoxyphenylhydrazone (FCCP) 1.5 µM, and Rotenone + Antimycin A 2 µM each. Spare respiratory capacity was measured as the change between OCR before and after FCCP addition. All measurements were normalized to cell count, obtained through crystal violet of seahorse plate immediately following seahorse assay.

### 2.13. Mitochondrial Reactive Oxygen Species

Mitochondrial ROS was determined in plate-reader format using mitoSOX red (ThermoFisher M36008). Cells were stained in Hank’s Balanced Salt Solution without phenol red, with 5 µM mitoSOX for 25 min. For positive control wells, 10 µM Antimycin A was added 10 min prior to the removal of mitoSOX staining. Cells were washed 1× PBS and read at 510/580 using a Biotek plate reader, then fixed for crystal violet staining. Fluorescent intensity was normalized to unstained control wells and cell count via crystal violet.

### 2.14. Bioinformatics and Pathway Analysis

#### 2.14.1. miR200c Gene Array Pathway Analysis

Data [Gene expression omnibus: GSE108271] were analyzed by one-way ANOVA performed at the gene and probe (exon) levels using Partek Genomics Suite software (Partek; St. Louis, MO, USA). Microarray data were normalized with a robust multiarray average method using Affymetrix Power Tools. Multiple probe sets for the same gene were collapsed using average expression. Genes with a false-discovery rate of <10% and a fold change of >1.2 were selected as differentially expressed between the two groups. Pathway analysis was performed using Gene Set Enrichment Analysis (GSEA) software and KEGG gene sets. Gene sets with *p* < 0.05 (after 1000 gene set permutations) were deemed to be enriched in each group. Metabolomics graphing was created with ggplot2.

#### 2.14.2. Auxotrophy Correlation

After 7 days of culture, the viability of cells in DMEM+5% LPDS was compared to DMEM+5% LPDS+ 10 µg/mL LDL. The relative change in growth (LDL/LPDS) was considered the “viability score”, and correlated with gene level data provided by the Cancer Cell Line Encyclopedia using spearman’s correlation coefficient. Genes were then ranked by *p*-value and spearman’s correlation. Those with a *p* < 0.05 were selected and ran in a GSEA pre-ranked analysis with 1000 permutations. Gene-level data for the top 4 genes were pulled and were reported with Pearson’s correlation coefficient.

## 3. Results

### 3.1. NPC1 Is Significantly Elevated in TNBC and Is Directly Targeted by miR-200c

Previously published datasets by the Richer lab demonstrated altered metabolism when miR-200c is restored to TNBC cell lines [12]. Pathway analysis of this dataset suggested lipid and cholesterol metabolism as potential targets of miR-200c (Appendix A), with *NPC1* as one of the most strongly suppressed metabolism-related genes (Appendix A). Functionally confirming this gene array data, exogenous restoration of miR-200c suppressed NPC1 at the mRNA and protein level in multiple TNBC cell lines (Figure 1A–C). The 3’UTR of *NPC1* contains a predicted binding site for miR-200c (targetscan.org) [20]; therefore, we tested for evidence of direct binding and regulatory activity at this site. Luciferase reporter assays confirmed direct binding of miR-200c to the wildtype, but not the mutated NPC1 3’UTR predicted binding sequence (Figure 1D).

The regulation of NPC1 in cancer has not been fully elucidated, but in normal physiology, NPC1 is induced through multiple transcription factors, including cholesterol-mediated sterol response element binding proteins (SREBPs) [21,22], and the cAMP response element binding protein (CREB) in steroidogenic cells [23]. As in normal glandular epithelial cells [22], we observed that TNBC cells increased NPC1 in response to low cholesterol (lipoprotein-depleted serum, LPDS) and serum-free media (SFM) conditions (Figure 1E). Although miR-200c attenuates total levels of NPC1 at baseline, it was unable to completely suppress NPC1 induction in low-serum and low-sterol conditions (Figure 1F; Appendix A), demonstrating that miR-200c influences, but does not independently control NPC1. Notably, miR-200c did not affect the low-density lipoprotein receptor (LDLR), the primary mediator of cholesterol uptake in epithelial cells [14] (Appendix A).

Supporting the relevance of NPC1 in breast cancer, a query of *NPC1* in publicly available patient datasets demonstrated higher *NPC1* mRNA in patients with basal, ER−, and high-grade tumors relative to ER+ low-grade disease (Figure 1G; Appendix A). By western blot, TNBC cell lines had higher expression of NPC1 relative to ER+ breast cancer cell lines (Figure 1H; Appendix A), and immunohistochemistry demonstrated NPC1 expression in both primary tumor and lung metastasis in an MMTV-driven mouse model of breast cancer (Appendix A). However, expression of *NPC1* in breast cancer datasets does not strongly correlate with genes in these pathways or the pathways themselves (Appendix A), suggesting that NPC1 is not upregulated due to broadly increased cholesterol or lysosomal signaling. 

These data revealed a novel miR-200c/NPC1 regulatory axis in breast cancer that may contribute to the differences in patient tumors observed in publicly available datasets, and suggest basal NPC1 upregulation as a unique feature in these tumor cells.

### 3.2. NPC1 Supports Breast Cancer Cell Invasion and Growth in Soft Agar

Based on the expression of NPC1 in breast cancer and its known roles in cholesterol metabolism and mTOR signaling pathways, we sought to determine if NPC1 supports tumor-promoting characteristics *in vitro*. Following shRNA knockdown of NPC1 using two unique shRNAs (Figure 2A; Appendix A), TNBC cells demonstrated characteristic “Niemann-Pick” punctate cholesterol localization [17,22] as visualized with filipin stain (Appendix A). These cells grew at a slowed rate in 2D culture compared to control cells (Figure 2B,C; Appendix A). In soft agar, which recapitulates anchorage-independent growth, NPC1 silencing decreased the number and size of colonies in Sum159PT cell lines (Figure 2D, Appendix A). TNBC cell lines were less invasive when NPC1 was silenced (Figure 2E). This is consistent with a recent study demonstrating that NPC1 inhibition reduces migration and invasion in Chinese hamster ovarian and A431 squamous carcinoma cell lines [24].

Strikingly, exogenous overexpression (oe) of NPC1 in ER+ MCF7 cells led to an increased number and size of colonies in soft agar (Figure 2F; Appendix A), and increased invasion of +NPC1 oe cells after 24 h (Figure 2G). The effect of NPC1 on anchorage-independent growth and invasiveness demonstrates regulation of multiple tumor cell-intrinsic properties relevant to the metastatic potential of TNBC. While NPC1 has not been explored in cancer, the literature on Niemann–Pick disease demonstrates the relevance of NPC1 in cholesterol and mitochondrial metabolism, which were evaluated next in the context of TNBC.

### 3.3. TNBC Cells Are Cholesterol Auxotrophs, but Do Not Solely Depend on NPC1 for Adequate Cholesterol Supply

In the absence of adequate LDL-derived cholesterol from the microenvironment, most cell types in normal physiology are able to produce cholesterol de novo through the cholesterol biosynthesis pathway [14,25,26]. However, recent studies identified “cholesterol auxotrophy”, the inability to survive without exogenous cholesterol, in varying types of cancers [27,28], suggesting cholesterol uptake as a targetable pathway. NPC1 specifically transports lysosomal cholesterol to the endoplasmic reticulum [29] where cellular cholesterol levels are sensed and controlled [17,30]. By depleting the endoplasmic reticulum of cholesterol, NPC1 inhibition affects both the supply of exogenous cholesterol as well, and disrupts cholesterol homeostasis [29,31]. How loss of NPC1 alters cholesterol homeostasis in BC has not been evaluated. Further, the dependence of breast cancer on exogenous cholesterol has not been reported.

To determine if disruption of cholesterol supply and homeostasis plays a role in the slowed growth observed in Figure 2, cholesterol auxotrophy in breast cancer was evaluated. A panel of breast cancer cell lines was grown in LPDS with or without supplemental LDL for 7 days, leading to significantly slowed growth in 7 of the 13 cell lines (Figure 3A). This occurred disproportionately in TNBC (5/6) as compared to non-TNBC (2/7). These trends were maintained during the 14-day culture (Appendix A). Of note, cholesterol auxotrophs grew normally for ~3–4 days, and then experienced a loss of viability over subsequent days (Appendix A). Since metabolic demands can differ in 2D versus 3D conditions, cholesterol dependence was tested over 14 days in soft agar, which demonstrated that cell lines maintain cholesterol dependence/independence in 3D anchorage-independent culture (Figure 3B).

In other models of cholesterol auxotrophy, which include ALK+ lymphomas and renal cell carcinomas, the phenotype was driven by low expression or mutations in key cholesterol biosynthesis genes [27]. However, loss-of-function mutations within these genes were not identified in breast cancer cell lines (Appendix A), and cholesterol auxotrophy does not correlate with mRNA levels of any cholesterol biosynthesis or uptake-related genes in these cell lines (Appendix A). While cholesterol auxotrophy weakly correlates (Pearson r = −0.63; *p* = 0.019) with NPC1 protein expression Fig metabolism (Appendix A). These data suggest that while cholesterol auxotrophy exists in a subset of breast cancer, it may not be driven by a single global mechanism. However, protein expression of NPC1 is positively correlated with dependency on exogenous cholesterol and may play a role in maintaining adequate cholesterol supply in cells that cannot produce it de novo.

To evaluate if NPC1 affects and supports cholesterol metabolism, key cholesterol readouts were evaluated following the NPC1 knockdown. Low cholesterol conditions drive activation of SREBP-1 and -2, which transcriptionally upregulate a large set of sterol-responsive enzymes. These include the low-density lipoprotein receptor (*LDLR*) and the rate-limiting step of cholesterol biosynthesis, HMG-CoA reductase (*HMGCR*) [32]. While cholesterol-related genes were differentially expressed in control versus shNPC1 cells, changes did not trend in one direction, nor were they consistent between cell lines (Appendix A). At the protein level, neither LDLR nor HMGCR were altered (Figure 3E) and the rate of LDL uptake was unchanged (Figure 3F), demonstrating that NPC1 loss does not specifically activate low-sterol responses in TNBC. 

It remained possible that loss of lysosome-ER cholesterol transport could result in a detrimental level of cholesterol depletion in NPC1 knockdown cells, given abnormal patterns of cholesterol localization in NPC disease [33]. To address this, soft agar growth media was supplemented with cholesterol complexed to Methyl-β-Cyclodextrin (MBCD), which delivered cholesterol independent of the lysosome [29,34]. However, MBCD:Cholesterol failed to rescue the growth of NPC1 knockdown cell lines (Figure 3G).

Together, these data demonstrate the novel finding of cholesterol auxotrophy in TNBC cancer cells, which correlates with NPC1 protein expression. However, NPC1 silencing did not cause dramatic changes to TNBC cholesterol metabolism readouts, nor did supplementation of cell-permeable cholesterol rescue cell growth. Thus, loss of NPC1-dependent cholesterol trafficking is likely not the mediator of the observed phenotype following NPC1 silencing in TNBC.

### 3.4. NPC1 Supports Mitochondrial Respiration and Maintains Mitochondrial Morphology

In models of Niemann–Pick disease, fibroblasts and stem cells demonstrated abnormal mitochondrial metabolism [16,35,36]. However, the mechanisms driving mitochondrial defects in this disease remain poorly understood. In TNBC cells, NPC1 silencing suppressed the oxygen consumption rate, a readout of mitochondrial activity, as evidenced by loss of basal respiration, and spare respiratory capacity (Figure 4A, Appendix A), while overexpression in ER+ MCF7 cells supported increased mitochondrial respiration (Figure 4B). Basal glycolysis was not affected by NPC1, but knockdown cell lines were less able to upregulate glycolysis following treatment by oligomycin, which inhibits mitochondrial ATP synthase (complex V). This suggests a limited ability to upregulate glycolysis to compensate for mitochondrial inhibition (Appendix A). Fitting with loss of mitochondrial respiration, NPC1 knockdown cells had increased basal levels of mitochondrial reactive oxygen species (ROS) (Figure 4D), and a greater relative increase in mitochondrial ROS when treated with Antimycin A, which is known to produce mitochondrial ROS due to electron transport chain complex III inhibition [37] (Appendix A).

NPC1 loss-of-function has also been associated with altered mitochondrial morphology [35], which could play a role in the observed decrease in the oxygen consumption rate, cell growth, and invasion of NPC1 knockdown cells. Indeed, NPC1 silencing in TNBC cells increased the percentage of cells exhibiting “short” and “intermediate” mitochondria while reducing the percentage of “elongated” mitochondria (Figure 4E,F; Appendix A). Further quantification with confocal microscopy demonstrated that while mitochondrial elongation was affected, the mitochondrial number was maintained (Figure 4G; Appendix A).

Mitochondrial morphology can be regulated by fission and fusion processes, however, no baseline changes to key fission/fusion proteins were detected. Interestingly, however, NPC1 expression altered fission/fusion signaling (MFN-1 and MFF) in response to metabolic and autophagic stress (Appendix A). Interestingly, mitochondria in a “donut” or “ring” morphology [38], a morphology unique from “shortened” mitochondria that resulted from fission processes, was observed in some shNPC1 cells. The donut versus short mitochondria is shown by electron microscopy in Figure 4H. Donut mitochondria are associated with decreased ATP production, defective calcium signaling, and increased mitochondrial ROS [39,40].

Together, these data demonstrate that NPC1 silencing impairs mitochondrial ATP production as evaluated by the oxygen consumption rate, which is associated with shortened mitochondrial morphology and increased mitochondrial reactive oxygen species (ROS). Given the key role of the mitochondrial function in cell growth and metastasis [41], defective mitochondria may contribute to the suppression of invasion and growth of TNBC cells following NPC1 inhibition.

### 3.5. NPC1 Affects mTOR and Autophagy Signaling under Stress Conditions in TNBC

NPC1 acts as the cholesterol “sensor” for the mammalian target of rapamycin (mTOR), the central regulator of metabolic-related proliferative signaling. In normal physiology and under cholesterol-replete conditions, NPC1 recruits mTOR to the lysosomal surface, where mTOR signaling is initiated [19]. In the absence of cholesterol, the altered conformation of NPC1 prevents this recruitment, prohibiting mTOR signaling [18]. In normal fibroblasts, neurons, and stem cells, the silencing of NPC1 allows constitutive activation of mTOR even under cholesterol-depleted conditions. However, mTOR signaling in normal nutrient conditions is not affected by NPC1 silencing in these cell types [18,35]. 

mTOR/AKT is activated in numerous cancers, generally through mutations in upstream regulators, including PTEN and PIK3CA [42]. Given the role of mTOR in cancer and its physical interaction with the lysosome and NPC1 [18,19,35], the western blot was used to evaluate how NPC1 silencing affects mTOR signaling.

In Sum159PT cells, loss of NPC1 caused a reduction in the mTOR target pS6K (Figure 5A). This suggests a regulatory role for NPC1 in the context of TNBC, unique from that in other published models [18,35]. To establish if NPC1 still serves as a cholesterol “sensor” for mTOR in cancer models, control versus shNPC1 cells were starved of cholesterol using methyl-beta-cyclodextrin (Figure 5A). As has been published in HEK293T cells [35], cholesterol starvation suppressed pS6K in control cells, but not shNPC1 cells, demonstrating that in these contexts, mTOR is sensitive to low cholesterol and depends on NPC1 to sense cholesterol levels. 

To further evaluate the effect of NPC1 silencing on mTOR, Sum159PT cells were treated in lysosomal and metabolic stress conditions for 24 h. These included lipoprotein depletion, amino acid starvation (AAS), and the lysosomal inhibitor bafilomycin A1 (BafA1). Knockdown of NPC1 caused a reduction in p-S6K1 and p-S6 in full serum and nutrient conditions (Figure 5B). Under nutrient and lysosomal stress, S6K signaling remained roughly the same between control and knockdown cells. However, nutrient deprivation coupled with NPC1 silencing caused further reduction in 4EBP-1 signaling compared to nutrient deprivation alone (Appendix A). Together, these data suggest that NPC1 silencing affects S6K versus 4EBP1 nodes of mTOR differently depending on nutrient status.

The effects of NPC1 on lysosomal metabolism and mTOR signaling implicates autophagy in this model, and autophagy defects have been noted in Niemann–Pick disease models. Indeed, NPC1 silencing causes increased LC3-II at baseline and in response to metabolic or lysosomal stress (Figure 5C), suggesting either increased production of autophagosomes or the buildup of autophagosomes. p62, which accumulates when autophagic clearance is impaired, was also increased in NPC1 knockdown cells. Together, these data were consistent with Niemann–Pick models where autophagosomes can fuse with the lysosomes, but cannot be properly degraded.

Together, these data demonstrate that NPC1 silencing acts on the mTOR pathway in association with decreased cancer cell viability, proliferation, and invasive capacity. Further, mitochondrial defects accompany this overall phenotype, and cells become less able to survive clinically relevant anti-cancer drugs.

### 3.6. NPC1 Has Therapeutic Potential Alone and in Combination with Chemotherapy in TNBC

We have shown that NPC1 silencing in TNBC causes mitochondrial dysfunction, decreased proliferation and invasion, and suppression of mTOR signaling (Figure 6A). Together, this suggests NPC1 as a potential target in this cancer subtype. Specific NPC1 inhibitors are not commercially available, although several small-molecule compounds have been shown to directly bind and inhibit NPC1 in addition to other targets with sterol-sensing domains [43,44], most notably, U18666A. To evaluate U18666A as a potential therapeutic, two TNBC cell lines were treated with serial dilutions of the drug for 48 h, which caused reduced cell viability at single-digit micromolar doses, while the ER+ cell lines T47D (Figure 6B) and MCF7 (Appendix A) were relatively insensitive.

U18666A was combined with paclitaxel because chemotherapy remains the frontline therapeutic in TNBC. This led to an additive or synergistic effect [45], depending on dose combinations (Figure 6C; Appendix A). These data demonstrate the potential utility of targeting NPC1 alone or in combination with existing TNBC therapies.

## 4. Discussion

TNBC is an aggressive breast cancer subtype with limited therapeutic options. The EMT-suppressor miR-200c is lost due to silencing or deletion in TNBC, allowing a phenotypic switch that supports invasion and metastasis [46]. Using miR-200c as a tool to identify novel pathways upregulated in TNBC, we found that miR-200c directly regulates *NPC1*, a lysosomal cholesterol transporter that we found to be elevated in TNBC compared to ER-positive disease. Literature evaluating NPC1 in cancer is extremely limited, with few studies showing elevated NPC1 in certain cancers, and implicating NPC1 in cell invasion and cell growth [24,47,48]. In our study, the knockdown of NPC1 in three TNBC cell lines led to a significant loss of proliferation in 2D culture and colony formation in soft agar. While we were unable to identify the precise mechanisms driving impaired proliferation and invasive capacity, the complexity of the NPC1 function in normal physiology suggested numerous mechanisms by which NPC1 could provide a survival advantage for TNBC.

Since NPC1 is a major provider of exogenous cholesterol to the endoplasmic reticulum, we hypothesized that disrupting this cholesterol axis could prevent cells from obtaining adequate exogenous cholesterol. A decreased growth in LPDS conditions existed within a subset of breast cancer cell lines and occurred more frequently in TNBC cell lines as compared to those representing other breast cancer subtypes. A demand for exogenous cholesterol weakly, but significantly correlates with the NPC1 protein expression, but not mRNA levels of other cholesterol-related genes. This suggests that while cholesterol auxotrophy exists within breast cancer, it may not be driven by a specific mechanism that is shared among all breast cancer auxotrophs. 

It is possible that cholesterol auxotrophic breast cancers have elevated NPC1 protein as a mechanism to support exogenous cholesterol availability. However, cell-permeable cholesterol did not rescue the viability of NPC1 knockdown TNBC cells in soft-agar conditions, suggesting that the slowed growth in these cells is not primarily driven by the loss of lysosomal-endoplasmic reticulum cholesterol transport. Further, TNBC cells did not exhibit an induction of sterol-related genes that generally respond to low-cholesterol levels, suggesting that cells received adequate intracellular cholesterol without the contribution of NPC1 in 2D and 3D cultures. This is likely due to the existence of unique transporters that transfer cholesterol to other organelles, such as the lysosome-mitochondrial transporter STARD3, which is implicated in Niemann–Pick disease [49]. Therefore, while NPC1 loss causes accumulation of cholesterol within the lysosomes, which is confirmed by the immunocytochemistry of cholesterol/filipin staining, it does not completely block access of all organelles to the lysosomal cholesterol pools.

Having ruled out cholesterol depletion as a primary mediator of the effects of NPC1 silencing growth inhibition, we evaluated mitochondrial metabolism. The NPC1 depletion led to a loss of mitochondrial respiration and an increase in mitochondrial ROS, which was associated with shortened and rounded mitochondria. The morphology of mitochondria is largely dictated by fission and fusion processes [50]. In the context of cancer, increased fission is believed to aid in invasion and migration, as shortened mitochondria can be more easily trafficked to the leading edge, and to areas of high energy demand [51,52]. This is inconsistent with the finding that NPC1 knockdown cells had decreased invasive capacity along with shortened mitochondria. However, we observed numerous “donut” mitochondria in NPC1 cells, which are believed to be caused by abnormal mitochondrial calcium signaling or ROS levels and have been noted to be less efficient in ATP production [53]. This suggests that the shortening of mitochondria in these cells may be driven by mitochondrial stress, and that these shorter defective mitochondria are not able to support the high energetic demands of tumor cell invasion. A limitation of our study is that we do not know the precise mechanisms driving altered mitochondrial morphology and function downstream of NPC1 in TNBC, which warrants future investigation. Interestingly, mitochondrial cholesterol content was increased in some Niemann–Pick models [54,55], while in others, mitochondrial biogenesis was suppressed [36] or mitophagy was defective [35]. Our novel finding that NPC1 silencing can suppress P70-S6K signaling in some breast cancer cell lines suggests that mTOR may link lysosomal cholesterol with mitochondria in these cells. 

Together, these data, characterized for the first time the effect of NPC1 inhibition in TNBC, and proposed a role for this protein in cancer. NPC1 expression was elevated in TNBC, and genetic inhibition of NPC1 caused decreased cell proliferation, growth on soft agar, and decreased invasive capacity. Based on the known roles of NPC1 in the Niemann–Pick disease, there are numerous mechanisms through which impaired growth and invasion may occur. We note that NPC1 silencing is associated with mitochondrial defects and suppressed mTOR signaling. Whether either of these defects is specifically directly causative of the observed growth and invasion, phenotypes remain to be established. We also found that U18666A, a small molecule that inhibits NPC1 activity, had a low micromolar IC50 against TNBC cell lines, as well as synergy in combination with the clinically approved chemotherapeutic, paclitaxel. Thus, targeting NPC1 could be of therapeutic interest, alone or in combination with standard treatments of TNBC.

## 5. Conclusions

TNBC is an aggressive disease subtype with limited targeted therapeutic options. In breast cancer, the loss of miR-200c promotes increased NPC1 expression, and NPC1 correlates with ER− and higher-grade disease. The silencing of NPC1 had significant effects on the proliferation and invasive capacity of TNBC cells, while overexpression of NPC1 in an ER+ cell line enhanced growth and invasion. We found NPC1 to interact with mTOR and mitochondrial function, however, the mechanisms specifically mediating the pro-tumorigenic effects of NPC1 remain to be established. Together, these data suggest that NPC1 is a potential target in TNBC. 

## Figures and Tables

**Figure 1 cancers-14-03543-f001:**
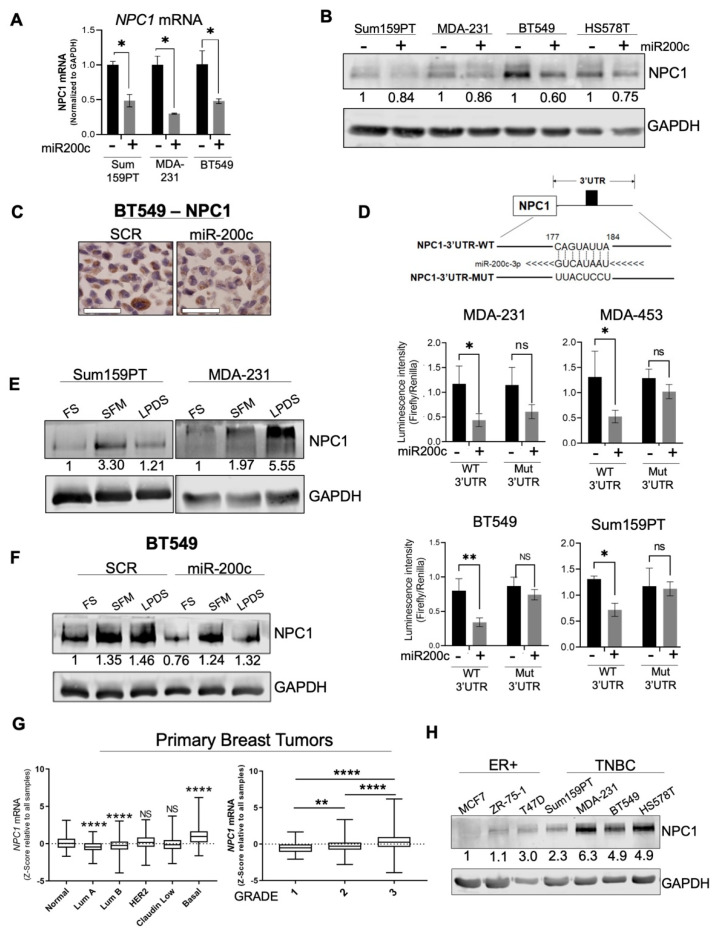
NPC1 is significantly elevated in aggressive breast cancers and is directly targeted by miR-200c. (**A**) Effect of miR-200c on *NPC1* in three TNBC cell lines by qPCR after a 48-h transfection. Unpaired *t*-Test (**B**) Effect of miR-200c on NPC1 following a 72-h transfection. (**C**) Immunohistochemistry of NPC1 in BT549 (/+ miR-200c, scale bar = 50 µm. (**D**) Luciferase assay performed on wildtype (WT) or mutated (Mut) *NPC1* 3’UTR, with and without exogenous miR-200c. (Student’s *t*-Test) (**E**) Western blot of NPC1 in serum-free media (SFM) or lipoprotein depleted serum (LPDS) for 24 h, in two TNBC cell lines. (**F**) Effect of miR-200c on NPC1 induction by a 24-h SFM and LPDS treatment. miR-200c transfection = 96 h. (**G**) *NPC1* in 2019 METABRIC breast cancer cohort by subtype (left, One-way ANOVA with Tukey Test, relative to “normal”) and grade (right, one-way ANOVA with Tukey Test). (**H**) Western blot of baseline NPC1, a panel of breast cancer cell lines. *p*-values denoted by asterisks where * ≤ 0.05; ** ≤ 0.01; **** ≤ 0.0001; or ns = not significant (>0.05).

**Figure 2 cancers-14-03543-f002:**
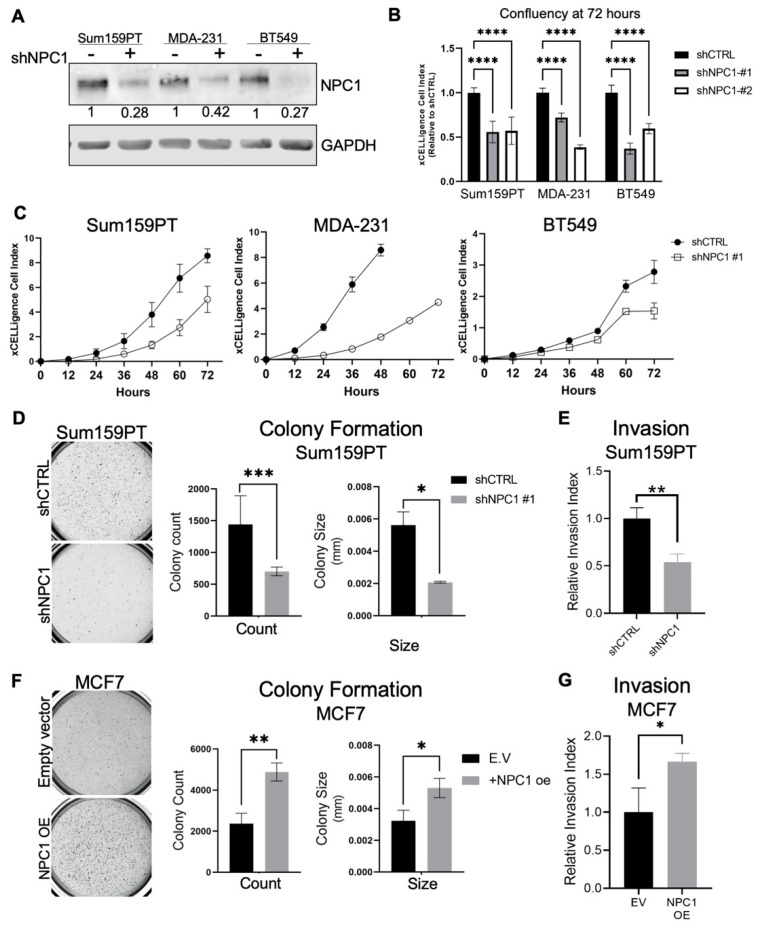
NPC1 supports breast cancer cell invasion and growth in soft agar. (**A**) Western blot confirmation of NPC1 knockdown by shRNA. NPC1 normalized to GAPDH. (**B**) Relative confluency of three TNBC cell lines with two unique shRNAs, following 72 h in 2D cultured, as measured by xCELLigence. (**C**) Growth of cell lines over 72 h, as measured by xCELLigence. (**D**) Sum159PT colony formation in soft agar, two weeks after seeding 3,000 cells/well. (**E**) Invasion of TNBC cells through Cultrex after 24 h, from serum-free DMEM toward 10% DMEM. (**F**) MCF7 colony formation in soft agar, two weeks after seeding 35,000 cells/well. (**G**) MCF7 invasion through Cultrex gel over 24 h. MCF7 cells transfected with an empty vector (pcDNA3.1) or constitutively active NPC1 (pcDNA3.1-NPC1). All statistics: student’s unpaired *t*-Test. *p*-values denoted by asterisks where * ≤ 0.05; ** ≤ 0.01; *** ≤ 0.001; **** ≤ 0.0001.

**Figure 3 cancers-14-03543-f003:**
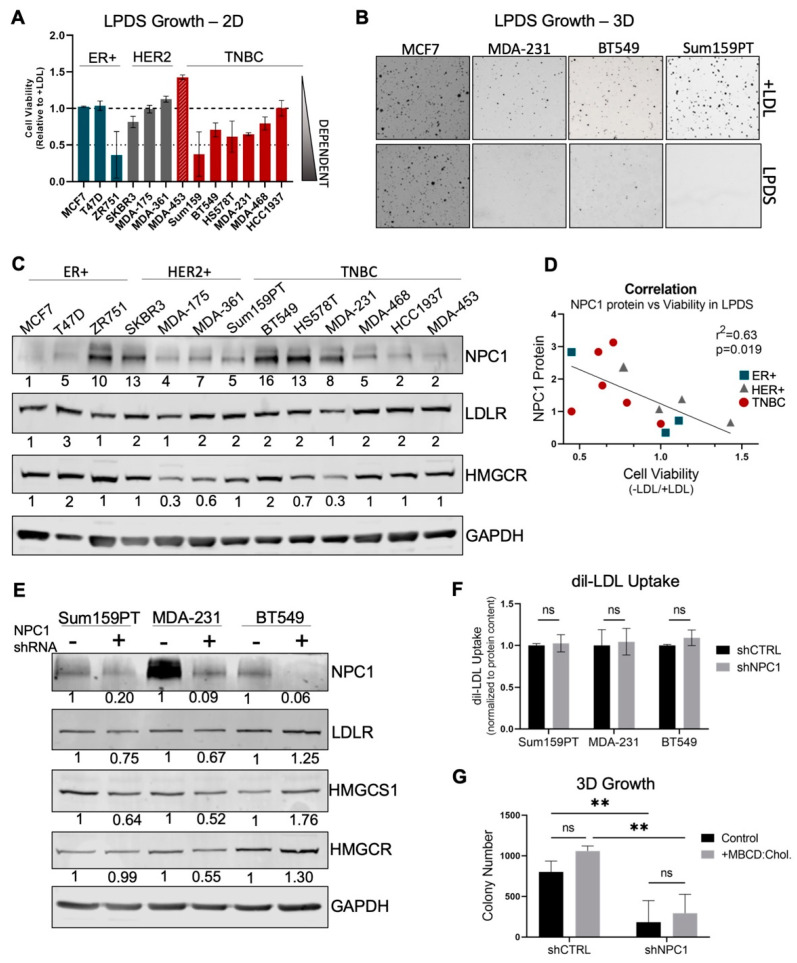
TNBC cells are cholesterol auxotrophs, but do not require NPC1 for adequate cholesterol supply. (**A**) Growth of breast cancer cell lines over 7 days in cholesterol-deplete (5% LPDS) relative to cholesterol-replete (5% LPDS + 10 µg/mL LDL) media. (**B**) Colony formation in soft agar, cholesterol-deplete (*top*) and –replete (*bottom*, 10 µg/mL LDL); fourteen days. (**C**) Basal levels of key cholesterol metabolism proteins NPC1, LDLR, and HMGCR in a panel of breast cancer cells. (**D**) Correlation of NPC1 protein levels (quantified from western, 3C) to cell viability in LPDS (viability score from 3A). (**E**) Western blot of key cholesterol proteins in shRNA#1 cell lines. (**F**) Uptake of fluorescent dil-LDL following 6 h of incubation, analyzed by a plate reader and normalized to protein content. (**G**) Growth of Sum159PT cells in soft agar when supplemented with MBCD:Cholesterol (1% MβCD complexed with 10 µg/mL cholesterol) over 14 days (Two-way ANOVA, Tukey Test). *p*-values denoted by asterisks where ** ≤ 0.01; or ns = not significant (>0.05).

**Figure 4 cancers-14-03543-f004:**
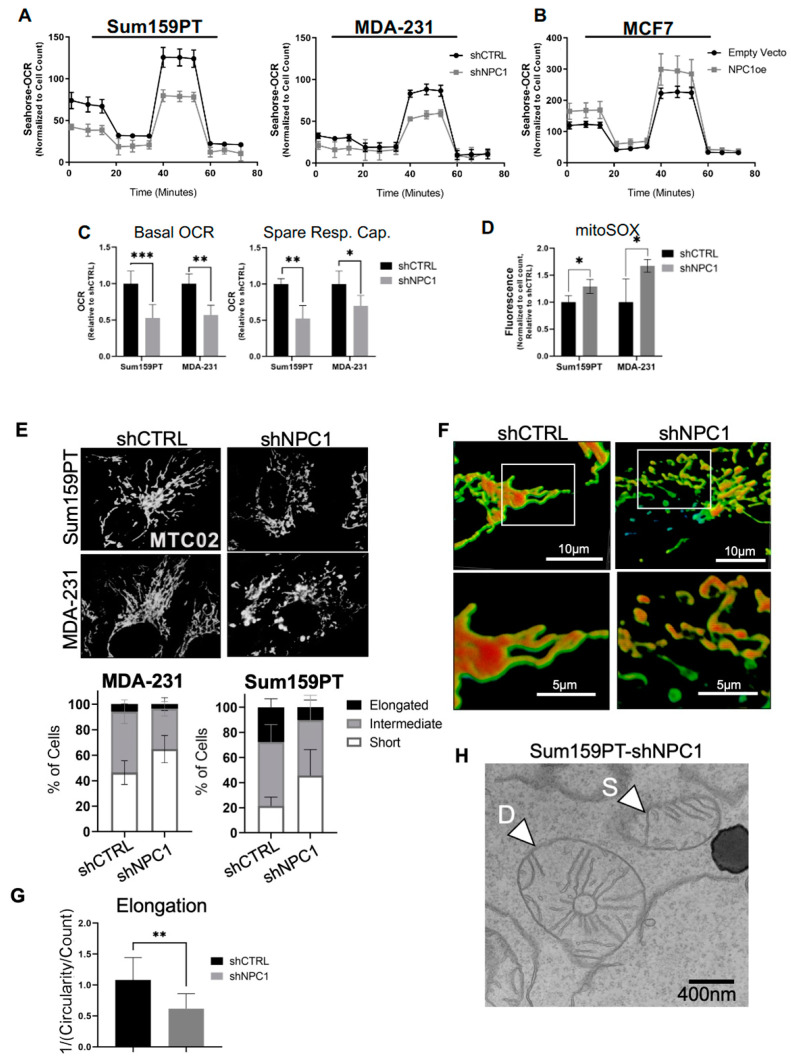
NPC1 supports mitochondrial respiration and maintains mitochondrial morphology. (**A**) Oxygen consumption rate (OCR) during Seahorse mitochondrial stress test, performed on Sum159PT and MDA-231 cell lines with NPC1 knockdown. (**B**) Mitochondrial stress test performed on MCF7 with pcDNA3.1-NPC1 or an empty vector. (**C**) Quantification of basal OCR and spare respiratory capacity of TNBC cells from mitochondrial stress test shown in A. (**D**) Mitochondrial ROS was evaluated using mitoSox staining. (**E**) *Top:* Representative images of fluorescence microscopy of TNBC cells stained with MTC02 (mitochondrial antibody). *Bottom:* quantification of cells with mitochondrial morphology defined as either elongated, intermediate, or short. (**F**) Confocal Z-stacking of mitochondria stained with MTC02, with representative images of ”elongated” (shCTRL) versus “shortened” (shNPC1) mitochondria. A snapshot of 60× with 4× zoom (top panel), and 60× with 8× zoom (bottom panel). (**G**) Quantification of mitochondrial elongation, as quantified by ImageJ. (**H**) Representative transmission electron microscopy image of short (“S”) and donut (“D”) mitochondria in Sum159PT with shNPC1 (30,000×). *p*-values denoted by asterisks where * ≤ 0.05; ** ≤ 0.01; *** ≤ 0.001.

**Figure 5 cancers-14-03543-f005:**
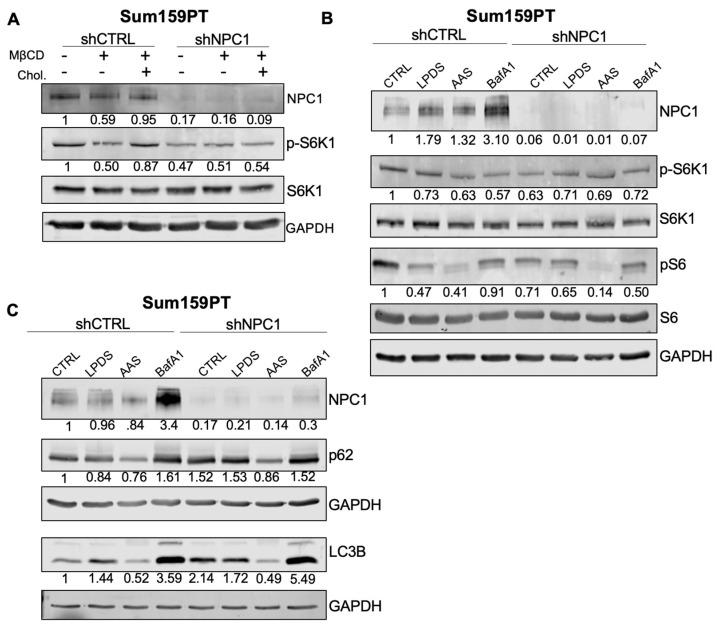
NPC1 affects mTOR and autophagy signaling under stress conditions in TNBC. (**A**) Western pospho-S6K (T389) signaling under cholesterol-stressed conditions. Control cells (lane 1, 4) were maintained in full serum. Where indicated, MβCD is used to deplete cholesterol for 2 h and then, as indicated (lane 3, 6), MβCD is removed and replaced with MβCD:Cholesterol to replenish cholesterol supply. Numbers below pS6K1 represent levels normalized to total S6K1. (**B**) S6 kinase signaling under metabolic or lysosome stress conditions for 24 h (LPDS−5% lipoprotein depleted serum; AAS-amino acid depletion with 5% FBS; BafA1-5nM). pS6K (Ser235/236) and pS6 are normalized to their total controls. NPC1 is normalized to GAPDH. (**C**) Autophagy markers p62 and LC3B under metabolic or lysosomal stress conditions.

**Figure 6 cancers-14-03543-f006:**
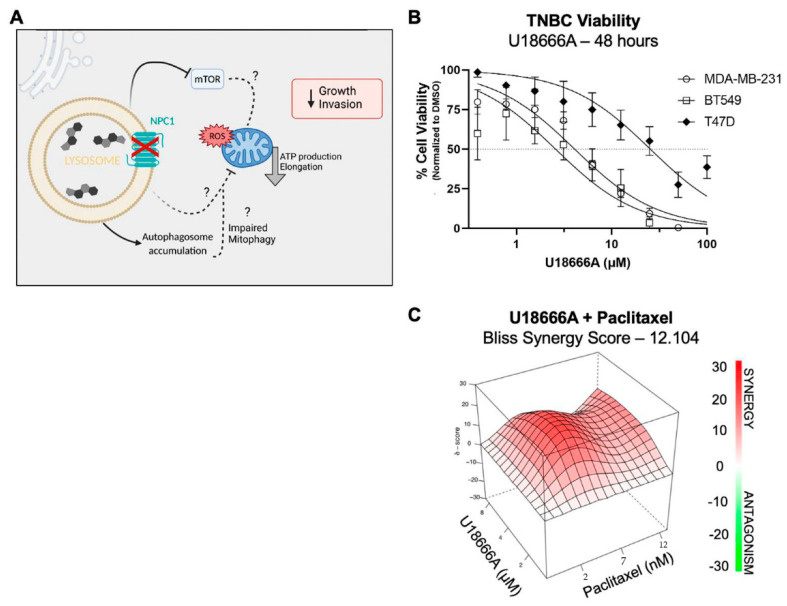
NPC1 has therapeutic potential alone and in combination with chemotherapies in TNBC. (**A**) A graphical summary of the anti-tumor effects of NPC1 silencing. Created with biorender.com (**B**) Dose-response curves of U18666A in TNBC (MDA-MB-231, BT549) and ER+ (T47D), 48 h drug treatment as analyzed by crystal violet. Linear regression with a variable slope. (**C**) Additive and synergistic effects of U18666A and Paclitaxel in BT549 cells. Three doses of each drug were given for 48 h and analyzed by crystal violet. The additive/synergy effect was analyzed and calculated using SynergyFinder 2.0.

## Data Availability

For any novel reagents or resources (e.g., modified cell lines or constructs) created in the course of this project, we will follow the NIH policy on the timely distribution and sharing of biomedical research resources, as published in the NIH Grants Policy Statement. As appropriate, sharing will be under the guidance of the CU Innovations technology transfer operations.

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
