# Peer review of "NPC1 Confers Metabolic Flexibility in Triple Negative Breast Cancer"

_cancers, 2022, doi:10.3390/cancers14143543_

Round 1

Reviewer 1 Report

The paper by o’Neill et al. shows that NPC1, (lysosomal cholesterol transporter Niemann-Pick Type C1) is highly expressed in triple negative cell (TNBC) lines compared to estrogen receptor-positive (ER+) breast cancer lines. NPC1 is directly targeted by microRNA-200c (miR-200c) a suppressor of EMT, this could explain its differential expression in breast cancer subtypes. NPC1 knock-out in TNBC causes an accumulation of cholesterol-filled lysosomes and seems decrease growth on soft agar and invasive capacity of these cells. In addition, NPC1 silencing alters mitochondrial function and morphology, inhibits mTOR signaling and induces autophagosome formation. These findings indicate NPC1 as a druggable target in TNBC.

This is an interesting paper describing a well-done work but, in my opinion, it is rather confusing. It is finally difficult establish the leading molecular mechanism by which NPC1 promotes aggressiveness and growth of TNBC.

Anyway, I think that the paper could be improved introducing a few control experiments:

First the Authors show that NPC is expressed at higher extent in TNBC compared to ER+ cells, but use only cell lines. I think that using of primary TNBC and ER+ cells would be much more informative.

In fig. 2 colony formation assay should be expressed non only as relative stimulation but also as absolute values and the effect of shNPC1 should probed even in MCF-7.

The Fig 4 could be better described, especially as concerns the panel 4H

In Fig. 6, the effect of U18666A should verified in MCF-7 too as a control.

Reviewer 2 Report

The study demonstrates the role of NPC1 in triple negative breast cancer. Niemann-Pick type C1 (NPC1) is an integral membrane protein on the limiting membrane of late endosome/lysosome that accept cholesterol from NPC2 and mediate cholesterol transport from LE/LY to endoplasmic reticulum (ER) and plasma membrane. The authors demonstrate high expression of NPC1 in TNBC cells directly targeted by miR200c. The authors have performed series of experiments to demonstrate the role of NPC 1 in TNBC and also establish this molecule as a therapeutic target. Overall, the studies performed are well conducted, and the results support the conclusions. There is a suggestion to provide a lay summary of the work and a schematic representation will be useful to the readers.

Round 2

Reviewer 1 Report

I think that most of my concerns have been addressed or discussed